# Methodologies for the *De novo* Discovery of Transposable Element Families

**DOI:** 10.3390/genes13040709

**Published:** 2022-04-17

**Authors:** Jessica M. Storer, Robert Hubley, Jeb Rosen, Arian F. A. Smit

**Affiliations:** Institute for Systems Biology, Seattle, WA 98109, USA; jessica.storer@isbscience.org (J.M.S.); robert.hubley@isbscience.org (R.H.); jeb.rosen@isbscience.org (J.R.)

**Keywords:** repeats, transposon, transposable element, *de novo* methods, signature-based methods, genome annotation, curation

## Abstract

The discovery and characterization of transposable element (TE) families are crucial tasks in the process of genome annotation. Careful curation of TE libraries for each organism is necessary as each has been exposed to a unique and often complex set of TE families. *De novo* methods have been developed; however, a fully automated and accurate approach to the development of complete libraries remains elusive. In this review, we cover established methods and recent developments in *de novo* TE analysis. We also present various methodologies used to assess these tools and discuss opportunities for further advancement of the field.

## 1. Introduction

Genomes have likely always battled with subsequences that evolved to multiply independently of genome replication. For billions of years, these transposable elements (TEs) have littered genomes with interspersed copies that are generally detrimental or useless for their hosts and thus tend to wither away over time. Depending on the relative rate of TE reproduction and genomic clean-up through random deletions, significant fractions of present genomes are ultimately derived from TEs. Recognized portions are as high as 84% in some cereals and 90% in lungfish [1,2,3]. Since 1980, it has been suggested that most of the 85–90% of our own genome that is not under functional constraint is TE derived [4]; and by 1996, we could confirm that for almost half the genome [5]. Because of relatively low TE activity and DNA loss, much of our and other vertebrate TE-derived DNA was introduced a long time ago and, through the accumulation of mutations, ranges from difficult to impossible to recognize as such. Over half of the human DNA recognizably derived from (~4 million) TE insertions became part of our genome over 80 million years ago, in a common ancestor of all placental mammals [6,7].

While the persistent onslaught of TEs has been a bane for genomes, as evidenced by the many and wide-ranging defense mechanisms they evolved against them, it forms a veritable boon for phylogenetic research. The advantages of TE insertions as a phylogenetic tool include their high abundance and interspersed distribution, the near-neutral nature of most insertions fixed in a population, the built-in knowledge of the ancestral (absent) state, the virtual absence of back-mutations or parallel events leading to the same sequence pattern (homoplasy), and our ability to recognize ancient events [8]. Not all TEs are equally suitable; less reliable are class II elements that excise from their locus during transposition or elements with more specific target site preferences. Most LINE elements, such as L1 in mammals and CR1 in birds, are close to ideal: random 5′ truncation of most insertions and the variable target site duplication (TSD) lengths distinguish even the rare event of same-site insertions in the same orientation in related genomes.

These qualities of TE insertions have been used with great aplomb to resolve long-standing phylogenetic problems before the availability of complete genome data [9,10,11,12,13]. Genome assemblies for most vertebrates and many other eukaryotes are being rapidly produced and can increasingly be studied in the context of complete and reliable multi-species genome alignments [14,15,16,17]. One could expect that the rich detail contained in whole-genome alignments would be more reliable in phylogenetic studies and that the role of TEs in phylogenetic studies will steadily decline, but they will always remain relevant in population studies and may even continue to be the best tool to resolve the trickiest phylogenies, such as species radiations. For these, individual markers show conflicting species trees because two or more speciation events took place when the loci were still polymorphic (incomplete lineage sorting) or due to interspecific gene flow (introgression), primarily via hybridization. Quartet-based summary coalescent methods have the potential to solve these knots when using TE insertions as input data [18,19]. For this to work, it is critical that the ancestral and derived loci are always correctly called, which is dependent on solid repeat annotation with full-length reconstructed TEs and knowledge of insertion behavior, and that the number of phylogenetic markers is high [20]. For phylogenetic purposes, the best product of *de novo* programs therefore is an as complete as possible library of reconstructed TEs, for anything but the youngest elements best presented by consensus sequences or profile HMMs.

TE libraries for the first sequenced genomes were years in the making and the need for automation was apparent early on, especially because the number of sequenced species was expected to grow exponentially. Indeed, in March 2021, the International Nucleotide Sequence Database Collaboration contained genome assemblies for 6480 unique species [21], including higher-quality assemblies for approximately 3300 animal and 800 plant species [22,23]. This is a small fraction of what awaits, with, for example, the Vertebrate Genomes Project aiming to generate complete reference genomes for all ~70,000 extant vertebrates [24] and the Darwin Tree of Life Project planning the same for all ~73,000 eukaryotic species in the UK [25]. These and many other such efforts are coordinated by the Earth BioGenome project to sequence all organisms in the forthcoming decade [21].

Earlier brute force TE library building efforts somewhat simplify repeat analysis in tetrapod genomes, as many ancient repeats are shared between these species and the general nature of the TE fauna is familiar. For most organisms, however, the vast majority of TE copies are lineage specific; a library has to be built from scratch and may contain heretofore unknown elements with their own idiosyncratic challenges.

Early in the 2000s, automation was addressed by a number of labs who developed programs such as RepeatFinder [26], REPuter [27], RECON [28], RepeatScout [29] and PILER [30] that are still in use today. We released an automated version of part of our own pipeline (RepeatModeler) in 2008 and new methods have been introduced steadily, to the point that prospective repeat analysts may be overwhelmed by the choices. Here, we provide an overview of the popular or promising new methods and pipelines to identify interspersed repeats *de novo*. We do not address the analysis of tandem repeats and satellites; the most recent reviews on this that we are aware of are from 2013 and 2015 [31,32] and quite a few promising newer methods have been published since [33,34,35,36,37]. The analysis of segmental duplications, which has seen considerable progress in recent years [38,39,40] also falls outside the scope of this review, though one should be aware of their existence as they can interfere with the discovery and analysis of TE families. We also do not address the genotyping of TE insertions compared to a reference genome (see reviews [41,42]) and more recently published tools [43,44,45].

Several reviews on the subject of *de novo* TE analysis have been written in the last dozen years [46,47,48,49,50]. We especially recommend the broader review by Nicolas and colleagues, originally written in 2016 and updated this year [51], which contains excellent introductions to the concepts of sequence indexing and their application to repeat detection. In addition, a comprehensive list of tools for TE analysis is currently being maintained as part of the TEHub project (http://tehub.org; accessed on 16 April 2022). Our focus will be on a comparison of the methodology of the most commonly used programs, the different ways the programs present the results, the need for a standard benchmark to meaningfully compare results of different programs, and some open problems that none of the programs have truly solved.

## 2. Why Is *De novo* Repeat Analysis So Hard?

At first thought, identification of interspersed repeats and subsequent calculation of a consensus sequence approximating the original TE appear straightforward. If most instances of TE copies have decayed in a neutral fashion, the accumulation of substitutions and indels should be random and with some knowledge of neutral mutation patterns, expectations can be set for what are likely dispersed copies of the same element instead of chance similarities. With enough copies, a reconstruction of the mobile element should be straightforward, but many complications exist.

While neutral decay provides advantages for repeat detection, the lack of selective constraint means that structural signals of TEs perish as quickly as any other sequence. Thus, a translational comparison or a search for characteristic terminal sequences does not increase sensitivity (quite the opposite).

The size of the genome can interfere with the detection of older and/or lower copy number elements. If TE instances have undergone 20% substitutions since arrival (an example of these are TE copies that arrived in the mouse genome at the time of speciation from hamsters), the distance between any two instances is on average 40%. Detecting matches of such a high divergence level requires very sensitive settings in self-comparison of the genome, making the process impractically slow. To allow more sensitive self-comparison, programs could work with smaller samples of a genome, but lower copy number elements may then go unnoticed.

Extensive fragmentation creates a challenge for algorithms to find the true ends of the TE. Older TE instances tend to be highly fragmented, either through partial deletions or through interruption by insertions, usually of other TEs. In many species, TE copies mostly accumulate in defined heterochromatic, gene-poor or intergenic regions of the genome, in part because their impact is more likely to be neutral. In those regions, overall repeat density can approach 100% of densely nested TE insertions [52,53]. On top of this, some elements tend to be truncated upon insertion. This is particularly so for LINEs, a class of elements that make up the majority of repeats in many vertebrates. 

To make things worse, full-length insertions are less likely to occur or persist than fragments, impeding their reconstruction. This can happen during transposition: cut and paste DNA transposons with an internal deletion appear to have an advantage over full, coding elements, perhaps because the transposase has a better chance binding to both termini. The ratio of short elements over full copies can be very high, hampering reconstruction of the long element. Often, the activity of a DNA transposon is only evidenced by the presence of tiny elements with terminal inverted repeats (TIRs) [54,55]. Autonomous elements with long terminal repeats (LTRs) may be outnumbered by elements with a reduced internal sequence [56,57] and LINE elements sometimes give free rides to internal deletion products [58]. Long insertions are also more likely to be selectively disadvantageous to the genome. Severely truncated LINE insertions are thus more likely to be fixed. For the same reason, full-length LTR elements are often reduced to solo LTRs via LTR–LTR recombination and the internal sequences of many LTR elements remain unknown.

Whereas the original sequence of most class II elements can be precisely reconstructed, class I elements evolve in a genome and the differences between instances are a mixture of neutral mutations accumulated in the fixed copies and evolved changes in the source gene(s). A consensus sequence of such families may not match any state of the evolving TE precisely. Over time, class I TEs can change to the point that homology between old and young copies or between copies of two branches is obscure. *De novo* algorithms will not consider these dissimilar copies to represent the same TE family. This is usually preferable, as a consensus or profile HMM sequence model of these aligned yet dissimilar sequences could be meaningless. Instead, several models will be built that represent often partially and sometimes wholly overlapping sets of instances of the evolving TE. Proper clustering of the instances is complicated, even if sometimes aided by apparent bursts of activity of the TE, resulting in clear “subfamilies” of instances. None of the *de novo* programs currently attempt an automated subfamily analysis.

Regional homology can exist between otherwise unrelated TEs, further complicating defining the true edges of a TE as well as its classification. These regional similarities have multiple origins. (1) A (fragment of a) TE could insert in or be recombined into an active other element. Some TEs, such as Helitrons and non-autonomous LTR elements, are particularly impartial to foreign intrusions. The anomalous L1-dependent SVA and LAVA elements active in ape genomes harbor *Alu* fragments, a retroviral LTR and a low complexity tandem repeat; had their copies been ancient and highly mutated, they would have been painful to reconstruct and classify. (2) Different elements may use the same functional module. The best examples of this phenomenon are probably SINEs. Classical SINEs originate by the happenstance recombination of a small structural RNA, containing an internal pol III promoter, and the 3′ end of a LINE element, which lets them hitch a ride with the latter. (3) Recombination between active TEs is a common feature. This is especially true for LTR elements, where disparate RNAs can be packaged in the same viral particle and template switching of the reverse transcriptase between the two RNA genomes is required for normal replication. Through this mechanism, chimeric-looking LTRs originate, identical LTRs can flank entirely different internal sequences (and vice versa) and integrases of one class of ERVs can even be combined with reverse transcriptases of another [59].

Besides these true recombinant mobile elements, *de novo* programs are also prone to build in silico chimeras of TEs, in part because integration site preferences of prolific TEs can make them frequently appear in tandem or at the (near) same site of other interspersed repeats. For instance, in mammals, L1-dependent SINEs insert in A-rich regions, most frequently provided by the poly-A tail of other SINEs. Because such incidental pairs can become a successful TE, the dimeric primate *Alus* being a prime example, they cannot be dismissed offhand.

Perhaps in part due to the frequency of tandem copies mentioned above, TE instances appear overrepresented in satellite-like tandem repeats [60]. Co-duplication of a TE instance such as that or along with segmental duplications can obfuscate its true extend. Given enough such copies and a significant decay from the original sequence, the model for the TE may become distorted as well. On top of that, *de novo* programs often build models of higher copy number segmental duplications or large tandem repeats as putative TE families. Being “random” fragments of the genome, these models may include coding regions of cellular genes and other unintended sequences. They usually contain copies of TEs and may prevent the discovery of some of these.

Until recently, known eukaryotic TEs ranged in size from 80 bp to approximately 15 kb. Unfortunately, oversized transposable units have now come to light in non-model organisms. These include members of known classes, such as 30 kb LTR elements in planarians, as well as exotic new TEs, such as the up to 180 kb Teratorns in fish [61]. These are a problem for TE class-specific programs, which necessarily impose size limitations, and tend to be fragmented by general *de novo* programs.

Finally, given the low information density present in a four-nucleotide alphabet, low complexity is the common source of false positives in the initial search for repetitive signals, false confidence in alignments of non-homologous sequences, and false extensions of real matches. This is of course particularly problematic for genomes with extreme GC content, but low-complexity problems can arise in subtler ways. *De novo*-created libraries almost always contain models representing common (degraded) simple repeats flanked by unrelated DNA of merely similar composition. Mini-satellites with the same periodicity and just a few bases in common will align “significantly” with each other in the long run and will show up as well. 

## 3. Approaches to TE Discovery and Annotation

The complex nature of TE sequence analysis is reflected in the often-ambiguous usage of terms such as “discovery” and “annotation”. It has been useful to define TE discovery as the process of reconstructing/modeling TE families directly from sequence data to generate or augment a TE library. Similarly, TE genome annotation can be viewed as the process of identifying and characterizing all recognizable instances of a TE family or a set of TE families in a genome. The large spectrum of methodologies developed for these tasks over the past two decades has blurred the lines between strict discovery and annotation processes. For this reason, we will focus on characterizing the granularity of results; from methods producing sequence ranges labeled simply as repetitive in nature to methods that produce complete family models and genome annotations. 

A further distinction is often made between methods which: (1) discover TE families based on general principles such as subsequence repetition/locality (*de novo*/ab initio methods); (2) employ domain knowledge to detect signatures of known TE class activity/composition (signature-based methods); and (3) methods that produce highly detailed genome annotation of TE instances and depend entirely on a predefined library of TE family models (library-based methods) (Figure 1). *De novo* and signature-based methods are typically employed to generate the input for library-based methods and will be the focus of this review.

The results of a *de novo* analysis come in a dizzying array of forms and granularity. Tools that provide genome annotations (marked with “Annotation Generation” in our software tables) may only separate a sequence into repetitive and non-repetitive subsequences, may report pairwise associations between repetitive subsequences, may group repetitive regions by broad TE classifications (e.g., LTRs and MITEs), or may rigorously report on distinct instances of clustered TE families. Tools falling into the last category typically also generate a library of sequence models for each family that has been discovered. These may take the form of consensus sequences, or representative instance(s) chosen from each family (exemplars). Furthermore, some programs provide provenance for the family definition in the form of sequence ranges for the identified family instances or a multiple sequence alignment of family instances (seed alignment).

## 4. *De novo* Methodologies

*De novo* methods have the advantage over signature-based methods that they can identify families that do not belong to a known class of TE or do not share one or more of the diagnostic features signature-based methods employ. They work by detecting exact or closely matching sequence repetitions, extending these matches, and in some cases grouping them into families of related sequences. In Table 1, *de novo* tools are characterized by the level of granularity they produce (families, instances, other), and by the types of family models produced (consensi, exemplars, or other tool-specific representations).

*De novo* methods that process whole-genome assemblies typically employ a self-comparison or (spaced) k-mer seeding approach. K-mer approaches identify over-represented exact *k*-length subsequences (k-mers) or *k*-length subsequences with a fixed pattern of match/mismatch positions (spaced k-mers) in an input sequence (Figure 2). These k-mer counts may be simply reported at every position (mer-engine), or thresholded to identify ranges of repetitive sequences (RAP, WindowMasker). The RED tool first identifies repetitive regions using spaced k-mer abundance, trains a classifier on these regions, and finally uses the classifier to annotate the genome. phRAIDER tiles spaced k-mers into maximal approximate matches (MAMs) identifying families of approximate repeats without indels. RepeatScout identifies abundant k-mers and employs them as seeds for a multiple sequence alignment extension and consensus generation. Finally, P-Clouds takes a statistical approach to cluster the k-mers with sequence overlap into groups (clouds). Regions showing significant coverage by k-mers present in one cloud are then considered repetitive and are annotated with that cloud.

Self-comparison approaches identify repetitive regions using computationally intensive alignment algorithms (Figure 3) followed by clustering strategies to resolve TE families from the pairwise alignment data. Accurate clustering of these alignments is challenging due to high fragmentation and mosaicism present in TE families. Grouper approaches this problem by applying single-linkage clustering, an agglomerative clustering technique that merges two clusters based on the shortest distance between any two members. RECON first applies single-linkage clustering, and then evaluates significant groupings of sequence endpoints within these clusters to identify composite sequences which are split apart accordingly. PILER uses several independent clustering approaches to identify tandem, local, and interspersed repeats from self-comparison alignment data. The interspersed repeat PILER method is also used in the CARP tool. It identifies clusters of aligned sequences that can be considered globally alignable. 

Many *de novo* methods have been developed that directly operate on next-generation sequencing (NGS) or single-molecule sequencing (SMS) reads. The primary advantage of this approach is to avoid assembler biases that often cause low-divergence repetitive sequences to be mis-assembled or left out entirely. The predominant approaches (Figure 4) attempt to treat repetitive sequence reads as a special case of sequence assembly or employ clustering methods to group reads/k-mers directly into repetitive families. In both cases, reads enriched for repetitive sequences are obtained either by downsampling the read dataset (to 0.1–0.5x coverage) or by filtering reads composed mostly of low-frequency k-mers. A few methods apply the assembly/clustering strategies directly to the k-mers rather than the reads. 

## 5. Signature-Based Methodologies

Purely *de novo* methods should be able to detect all classes of TE families. However, detecting TEs by sequence repetition alone has the potential to miss low-copy or certainly single-copy members of any well-characterized class of TEs and leads to the inclusion of non-TE sequences, such as processed pseudogenes and high-copy gene families. Without expectations regarding the structure of mobile elements, these methods also produce many fragmented or overextended TE models. LTR elements are particularly vulnerable to this, and the output of *de novo* programs may contain (fragments of) solo LTRs, single LTRs with a fragment of an internal sequence on either or both sides, all the way up to LTR-int-LTR-int-LTR structures. Signature-based methods are less susceptible to these particular problems.

Signature-based methods (Table 2) identify TE instances (Figure 5) by recognizing features of specific classes of TEs (terminal inverted repeats, direct repeats, transcription factor binding sites, protein motifs, etc.) as well as hallmarks of TE insertions, such as target site duplications. Often, several features must be used in concert to overcome the low specificity of each; even still, signature-based methods typically suffer from high false-positive rates. 

LTR retrotransposons and non-autonomous DNA transposons (aka MITEs when very short) are particularly suited to this approach due to the presence of long direct and inverted repeats flanking intact copies, respectively. Fast computational approaches have been developed to identify generic locally duplicated direct or inverted repeats, but this property alone is not sufficient to identify a TE instance. LTR and MITE finders use these methods to identify potential candidate matches, which are then further evaluated for the presence of target site duplications (short genomic sequences duplicated at the time of insertion), non-repetitive flanking sequences (e.g., not part of a larger repetitive element), and the presence of motifs/protein domains. 

## 6. TE Discovery Pipelines

A TE discovery pipeline is defined herein as a combination of previously published or discrete *de novo* algorithms to comprehensively describe all TE classes within any given genome. A pipeline represents a protocol for the orchestration of various tools and their parameters, often augmented with algorithms for clustering TE instances, defragmentation, sequence modeling, and the reduction in false positives and redundancy. The strategies of several popular pipelines are outlined in Figure 6, and further detailed in Appendix A.

Integrating a variety of tools into a single pipeline is an effective way to overcome the shortcomings of any one particular approach; however, it also introduces its own set of challenges. For instance, the outputs of different tools may substantially overlap with each other requiring complex adjudication strategies to eliminate the redundancy. This process is complicated by the natural fragmentation and high sequence divergence present in many TE instances. Managing the overall false-positive rate is a further challenge when integrating several discovery approaches, as each additional tool will contribute its own distinct set of false positives. Finally, each additional method requires evaluation of a larger set of possibly dependent parameters for the overall process.

Various strategies have been employed to produce either a library consisting of a unique set of TE family exemplars or consensi, or of distinct genomic instances. To that end, pipelines utilize clustering methods to collapse similar instances into families, or even redundant families into a single entry. One approach is to use fast sequence clustering algorithms such as CD-HIT-est for this purpose, efficiently grouping sequences with high sequence similarity (>75% sequence identity [102]) (PiRATE, RepeatModeler). A similar approach uses alignment tools (BlastN, Vmatch, etc.) to more accurately assess sequence similarity (albeit less efficiently) and cluster the pairwise sequence distances using single-linkage or complete linkage clustering techniques (EDTA, MAKER-P, tephra, REPET). In addition, pipelines that use sequential discovery and masking stages avoid the inter-tool clustering problem altogether (RepeatModeler). This is an area that is likely to see improvement in coming years as novel sequence distance estimation [103] and clustering techniques [104] are evaluated in the context of TE families.

The *de novo* tools invoked by a pipeline may have varying levels of false positives, including matches to coincidental groupings of low-specificity sequence signatures, inclusion of sequences in segmental duplications, low-complexity/tandem sequences, or identification of gene families. Combining the results of multiple tools compounds these problems. One approach to reduce false positives has been to consider the flanking or partially flanking regions of instances, filtering those that demonstrate a level of repetitiveness either genome-wide (evidence that either edge is part of a larger repeat), or between two edges (indicating that the extents of the repeat were not fully recognized) (EDTA, MAKER-P). Filtering such instances may be effective in reducing false positives but may also catch true instances. An approach that specifically targets false positives induced by low-complexity sequences and tandem repeats is to pre-mask these regions prior to running discovery tools and only restore them if they are found to be flanked by repetitive sequences (RepeatModeler, REPET).

*De novo* methods often produce family definitions representing mere fragments of a full-length family. In many cases, more than one fragment is present, representing different regions of the same TE family and creating a problem when they are not recognized as such. Family fragmentation produces inaccurate estimates of families and their abundance, often hampers correct classification of the family, and produces confusing nestings of annotation. However, popular pipelines have yet to tackle this crucial problem.

Since the final output of a TE discovery pipeline may consist of instances, exemplar sequences, and/or consensus sequences from a variety of underlying algorithms, it is important to be aware of the relative limitations of the different data types and appropriate uses. Given perfectly random, neutral decay, and a consensus sequence that precisely matches the ancestral TE, the substitution level of an instance from a consensus is twice as low as that from the average other instance. While this ideal situation is not always met, alignments against exemplars will give an average higher divergence of the TE family, resulting in a higher estimate of the age of the TEs. The sensitivity of the alignments is also reduced, drastically so when the actual average substitution level is over 15–20%. Some tools may provide outputs of the intermediate results or provide the multiple sequence alignments (seed alignments) for the derived TE family consensi. The latter provides a useful definition for the family and from which a consensus may be further improved or other forms of sequence modeling, such as profile Hidden Markov Modeling (pHMMs) may be applied.

The pipelines discussed here have been evaluated to varying degrees on non-model organisms, distantly related species to those used while developing the pipeline, or on species harboring differing TE content, which may represent a wider range of sequence divergence. This is often attributed to the different compositions of TE classes within different species. Therefore, the appropriate pipeline and associated strengths and weaknesses should be considered before beginning any genome analysis. Unfortunately, it can be difficult to make a direct comparison between even *de novo* tools, and much more so for pipelines. In particular, there is not a clear standardized or widely adopted benchmarking method for comparing the relative quality of libraries or genome annotations.

## 7. Benchmarking

The high diversity of benchmarking approaches applied to TE discovery is a barrier to both the understanding of the true performance of a method and to the competitive evaluation of methods. While this issue has been previously identified [105], a universal approach to this problem has yet to be developed. Of the many benchmarking methods, the comparison of results to existing highly curated libraries or genome annotations (i.e., gold standard dataset) has been the most frequently employed method for assessing true positives (TP) and false negatives (FN). The gold standard used is dependent upon the product and/or goal of the program in question. For example, if the products are consensi, these are typically compared to Repbase [106] or Dfam [107] consensi in their entirety or to a random sample. For algorithms targeting one type of TE (e.g., LTR elements), these elements are extracted from the Repbase consensi based on the criteria the authors have set and compared to the program output. Alternatively, two common options are utilized if genome annotation is the output and/or goal: (1) the generated library and the Repbase library are each used for RepeatMasker runs, and the loci compared or (2) loci obtained from previously published data are utilized for comparison. The main assumptions when using gold standards (e.g., Repbase, RepeatMasker annotations, or previously published data) are that these data are complete and accurate.

Similarly, the assessment of false positives (FP) is often accomplished by running a given method on a randomized, shuffled, masked or simulated genomic sequence in which any result is necessarily false. The simplest approaches, randomizing bases or shuffling words, do not maintain the complexity of the genome (e.g., maintaining isochores and commonly repeated k-mers) which can produce a less challenging benchmark. Masking out known elements has the advantage of preserving natural background sequences and other non-target genomic elements, but assumes that the masking process is ideal and no true copies of the target are present. Sequence simulation is an attempt to use sequence models, trained on the natural sequence, to generate sequences with similar complexity. The GARLIC [108] tool is an example of this last approach, in which a model trained on the genome is used to generate sequences with the addition of simple/tandem repeats to create a realistic background sequence. In addition, the inclusion of simulated instances generated from TE consensus sequences and fragmented/mutated to natural divergence levels allows the same benchmark to be used to comprehensively assess TP, FN and FP results. Simulation is particularly well suited to repetitiveness-based *de novo* algorithms, but may be less appropriate for programs that detect intra-TE signatures such as TSDs without extra treatment in the simulation. 

In addition to assessing their newly generated algorithm, authors may compare their approach to similar tools. In these cases, the most common metrics include a copy number comparison between genome annotations, and/or comparison of the number of models generated, the length of the sequences generated as part of the program output and the N50 of the library. Such metrics promote the idea that more is better. However, this notion does not take into account the quality of the dataset produced.

## 8. Cleaning up a TE Library

While *de novo* programs are sometimes used to directly annotate genomes, a careful comparison against a pipeline’s complete library or the combination of the output with previously established models has the advantage that the best match can be chosen between two or more related models. Other advantages of the use of libraries are reproducibility, provenance, and the possibility of incremental improvements. 

An ideal library would contain only full-length models of all significantly distinct TEs that have left copies in a genome. Full-length models would not only make annotations easier to interpret and allow reconstruction of evolutionary events, important for, e.g., phylogenetic analysis as pointed out in the introduction, but also avoid unfair competition between more or less complete models of related TEs. While such an ideal may never be reached, all automatically created TE libraries currently still need extensive editing before they can be accepted in curated databases such as Dfam or Repbase. The work involved is so intense that the great majority of Dfam submissions is currently housed in a non-curated section [109]. Libraries can always be improved, and many imperfections that seem hard to address automatically may be fixed in updated TE repositories after manual intervention. 

There are some significant common library deficiencies that pipelines can ameliorate with additional filters or modules. These include extensive redundancy, and the presence of false positives, artifacts and genic DNA. Below, we also briefly discuss the generation of relevant (sub)family models, but do not address complex problems such as filtering composite artifacts, identifying overextensions and finishing or merging fragmentary models. These are largely open problems for automated methodologies and still require extensive manual curation to identify and remedy.

While some pipelines (soft) mask simple repeats before *de novo* analysis, the output from all still includes many low complexity sequences with degenerate simple repeats at their cores. Entries that are almost entirely masked by a combination of low complexity and simple repeat finding programs could generally be dismissed automatically at the end of a pipeline. Even if some simple repeats, such as telomeres, are interesting to annotate, genome annotation programs tend to include a separate tandem repeat finding module that would identify these before comparison to the library.

The output of signature-based programs usually contains a number of false positives comprising random genomic sequence. Their uniqueness is a red flag noted by some pipelines, but should be weighed against the evidence, as low/single-copy active TEs are often of considerable interest. *De novo* programs also produce entries looking like random genomic DNA. These may be long 3′ UTRs of processed pseudogenes or, more often, fragments of segmental duplications. In seed alignments, they show a lack of defined ends and true TE instances within them show up as short, dispersed regions with a high number of seeds aligned. Lacking fully automated interpreters of seed alignments as yet, pipelines could instead mark entries as possible segmental duplications if they match multiple other library entries with various classification and if the genome annotation step, already part of most pipelines, shows them to be primarily localized in distribution.

Models that represent signature-based false positives or segmental duplications can contain coding regions of cellular genes. These are unwelcome in repeat libraries, if only for the natural tendency of most researchers to ignore DNA annotated as repetitive. All *de novo* methods also create (partial) models of highly expressed mRNAs from the interspersed processed pseudogenes in a genome, and often create models for common conserved or tandemly repeated protein domains, zinc-finger motifs being a stalwart. While some pipelines offer filtration of genic DNA, these rely on user-supplied protein libraries or repeat-free genomic DNA, the quality of which is critical and not easy to achieve. Furthermore, proper TE models may be dismissed by distant homologies to cellular proteins. Instead, pipelines could offer a competitive comparison to a curated TE protein database [110] and a domain database such as Pfam [111] from which TE protein domains have been removed. TEs may carry (fragments of) cellular genes along, so this filter should be conservative. 

Each discovery program has a tendency to produce redundant but non-identical entries for some TE families, especially when they are abundant. For a primate genome, for example, dozens of generic *Alu* models are built. A major drawback of using multiple programs through a pipeline is rediscovery of the same family, resulting in a dramatic increase in redundancy. This causes confusing genome annotation, with instances of the same TE receiving different labels, and more false-positive matches. This is not a small problem; in our experience automatically produced libraries are often more than twice too large. Most pipelines include a clustering step to reduce redundancy, but this usually only involves one class of elements. Defining a redundant set and choosing the best representatives is not a straightforward task, especially since many programs do not preserve the seeds and alignments that led to each model. Simple rules, such as merging all models for which the consensus sequences are 80% similar over 90% of the length, generally do not suffice. Under those restrictions, good models representing distinct subfamilies of a class I element may be lost, while bad models of the same TE (with significant errors, many ambiguous bases and/or long false extensions) will be retained. For *Alu*, the redundant models are often quite diverged, because the many, highly mutagenic CpG sites have been called differently to TG or CA in each. As many models are incomplete, two models with long non-aligned extensions on opposite sites may very well represent a single TE family but will not be joined following such rules.

On top of the problems of recognizing genuine duplicates, trying to define what constitutes a TE family using set cutoffs is impracticable. For phylogeneticists, recognition of the youngest, potentially dimorphic subfamilies of a TE is important. However, if all instances more than 80% similar over 80% of their length should constitute a single family and model [112], young branches of a long-term resident class I TE would go unnoticed as they will be absorbed by a general model. Additionally, still relatively young elements with a >10% substitution level, would never be built, since most or all of their copies are more than 20% diverged from each other. Generation of one model for many related TEs in general leads to an overestimate of the divergence and therefore age of the copies, which can confuse evolutionary analyses. For older TEs, it also leads to a significant loss in sensitivity during annotation.

Our hands-on strategy to reduce redundancy is to combine the seeds of all interrelated models and perform subfamily analysis using Coseg when all models involved align over much of their length, or create subfamily models with a CD-HIT based code when relationships are partial or modular [113,114,115]. Such a strategy could be incorporated in pipelines.

As curators of Dfam, we are also acutely aware of redundancy between libraries of related genomes. For example, most of the TEs described in the human genome have left copies that are (originally) shared between all placental mammals. The majority of a library constructed for a slow-evolving genome such as that of a rhinoceros will match those ancestral elements and would form redundant entries (as all ancestral models are or should be included in the analysis of a genome). Since very similar TEs have been active in separate lineages of mammals, sequence similarity is not the right basis for detecting this redundancy. Instead, observation of their presence at orthologous sites outside the lineage is key [116]. Fortunately, as mentioned in the introduction, in the near future, so many species will have been sequenced that most genomes can be studied in the context of multi-species genome alignments. In those settings, not only can it be quickly determined if TE copies are ancestral to two species, but one may avoid rediscovering ancestral TEs by focusing on those parts of a genome that are lineage specific. Many other advantages of such subtractive TE detection, which is as old as the publication of the first aligned mammalian genomes [117,118,119], suggest that this method may be the future of *de novo* TE detection.

## Figures and Tables

**Figure 1 genes-13-00709-f001:**
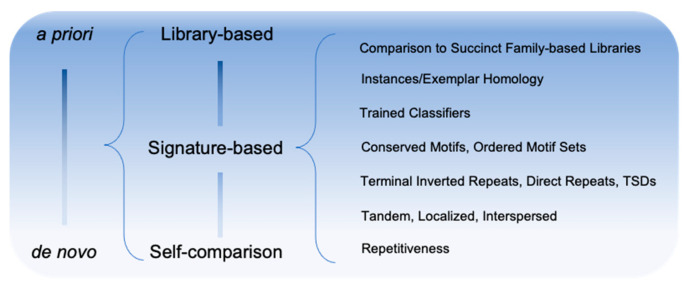
Spectrum of methodologies for the discovery of TE sequences.

**Figure 2 genes-13-00709-f002:**
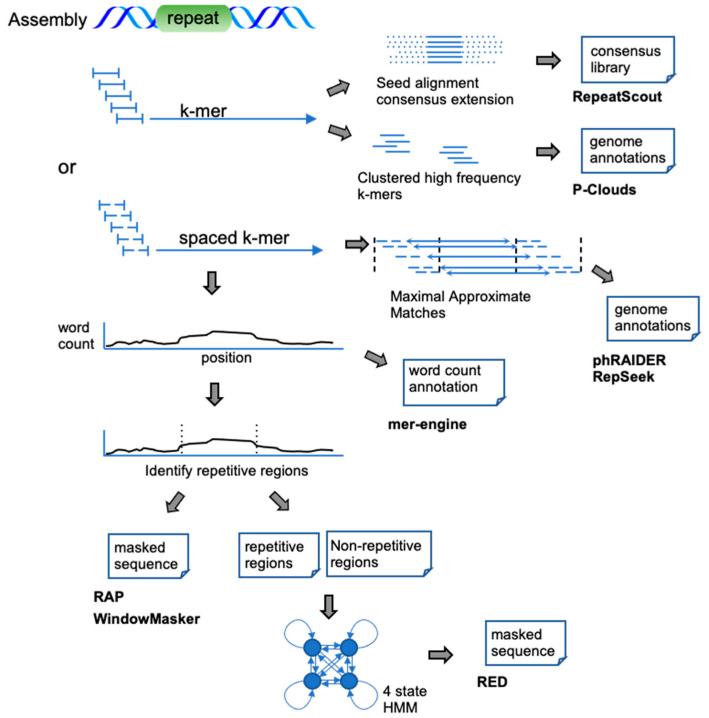
K-mer-based approaches on sequence assemblies. Upon characterizing the k-mer composition of the assembly, the word counts are either: simply used to annotate each base of the sequence (mer-engine), used to discriminate regions of high repetitiveness (RAP, WindowMasker, RED), clustered (P-Clouds), or used as anchors in a seed and extension process (RepeatScout, phRAIDER, RepSeek).

**Figure 3 genes-13-00709-f003:**
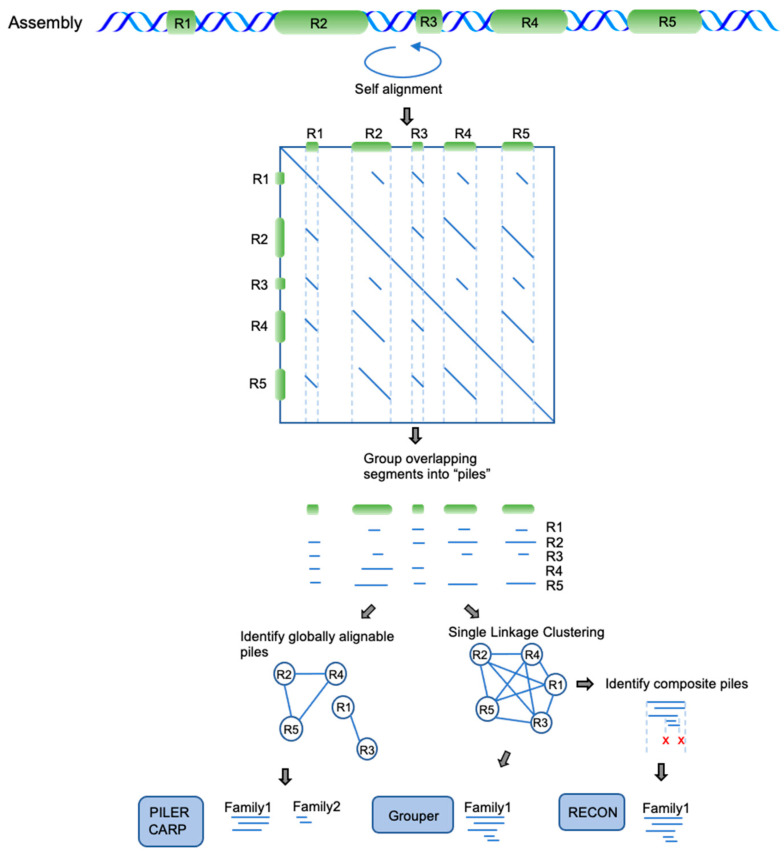
Self-comparison approaches. These methods attempt an all-vs.-all self-alignment using the whole assembly or a portion thereof. The self-alignments, viewed as a dot plot, will have many off-diagonal alignments representing dispersed similarities. These methods group the alignments into “piles”, defined by their distinct coverage across a region of the assembly. The primary difference between methods is in how they group piles into families. PILER and CARP require that elements are globally alignable, thereby identifying R1/R3 as a distinct family rather than fragments. Grouper and RECON apply single-linkage clustering, which, in this example, groups all fragments into a single family. RECON further attempts to identify composite families by looking for overrepresented internal edges—in this example, the internal edges were not deemed significant (red x’s).

**Figure 4 genes-13-00709-f004:**
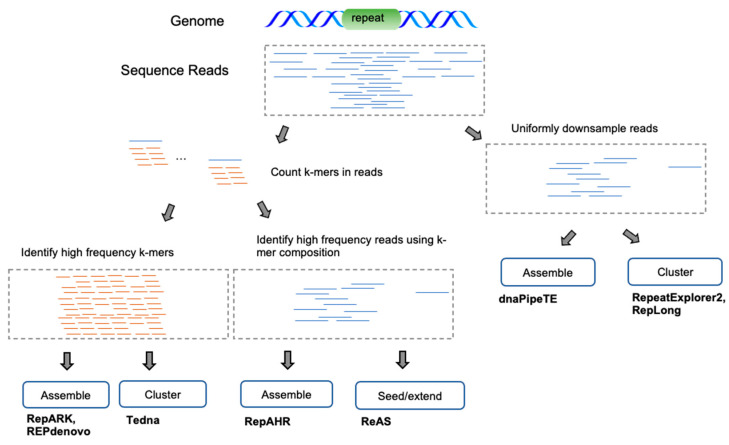
Read-based *de novo* methodologies. Due to the overwhelming size of read datasets, methods often start by either downsampling or filtering low-coverage regions based upon read k-mer frequencies. At this stage, either the remaining reads or the k-mers themselves are assembled into contigs or clustered into distinct groups representing repetitive families.

**Figure 5 genes-13-00709-f005:**
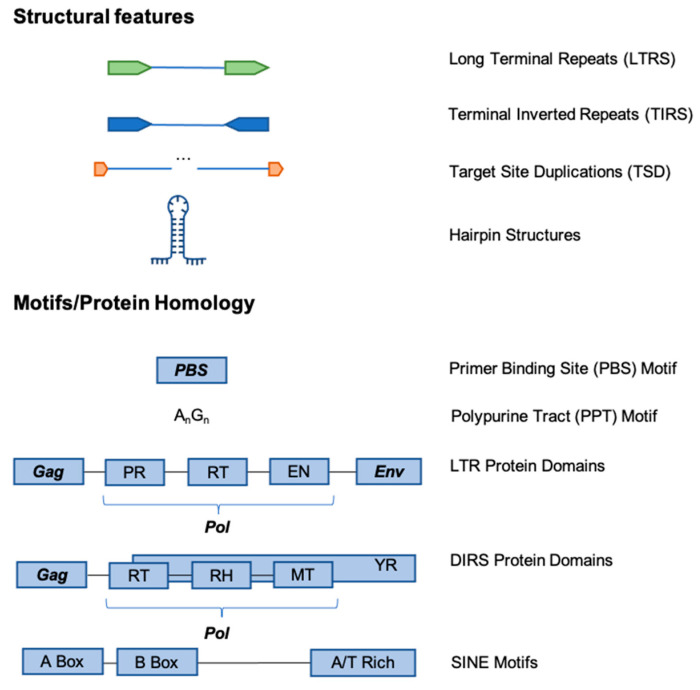
Examples of commonly used TE Signatures for Detection. Structural features: the identification of LTR/ERV elements, class II elements, non-LTR retrotransposable elements, and Helitrons can be achieved by searching for LTRs (~100–1000 bp direct repeats), TIRs (~10–40 bp inverted repeats), TSDs (6–21 bp on average duplications), and hairpin structures, respectively. In addition, the A and B boxes seen in RNA polymerase III promoters and 3′ terminal A/T-rich sequence can be used to identify SINE elements. Motifs/Protein Homology: the order, orientation, and similarity to protein domains is key to homology-based searches. Gag: group-specific antigen; PR: pathogenesis-related; RT: reverse transcriptase; EN: endonuclease; Env: envelope; RH: ribonuclease H; MT: methyltransferase; YR: tyrosine recombinase. Other sequence structures (not seen in the figure above) observed in LINEs are their poly-A or simple-repeat tails, and the RT and apurinic–apyrimidinic EN (APE) domains of the Pol protein.

**Figure 6 genes-13-00709-f006:**
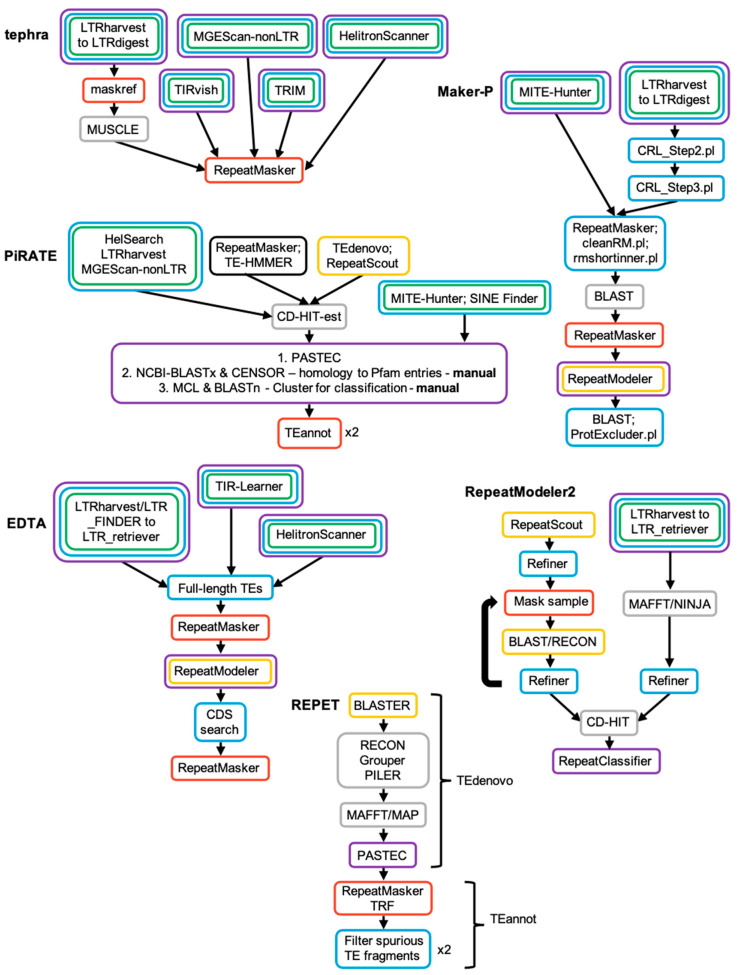
Workflow of select TE discovery pipelines. Each process in the pipeline has been categorized as classification (purple), signature-based TE detection (green), *de novo* TE detection (gold), homology-based detection (black), genome annotation (red), filter and/or refinement (blue) and clustering (grey). Arrows indicate the general workflow direction. NOTE: the above image is meant to describe the high-level organization of each pipeline, and does not reflect the inherent complexity contained within. Refer to Appendix A for additional details.

**Table 1 genes-13-00709-t001:**
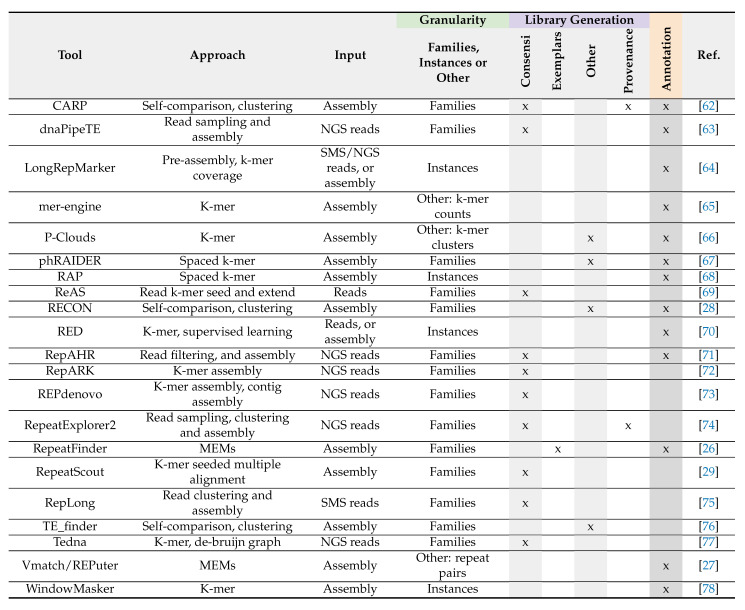
*De novo* methods.

**Table 2 genes-13-00709-t002:**
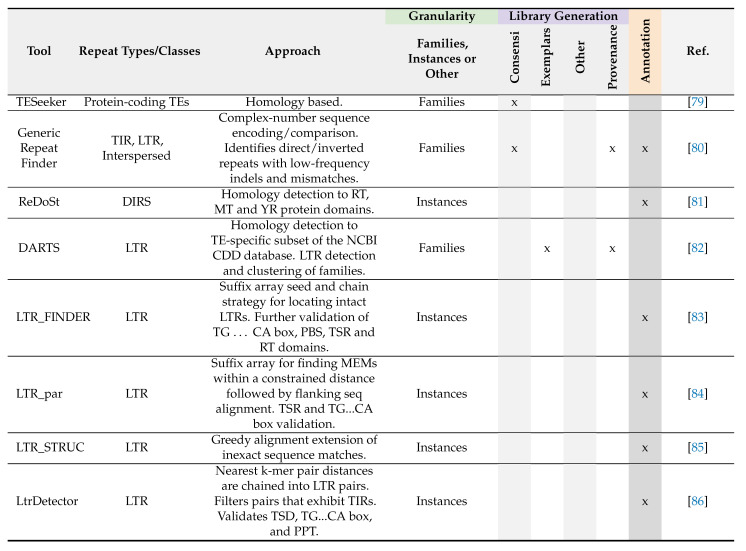
Signature-based methods.

## Data Availability

The Appendix A are available on the online version of this paper.

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
