# Peer review of "Methodologies for the De novo Discovery of Transposable Element Families"

_genes, 2022, doi:10.3390/genes13040709_

Round 1
Reviewer 1 Report
The authors mainly focused on the method of de novo annotating transposon. This review summarized and compared the relevant tools, pipelines, and methodological principles and provides a good summary of the important research content in the field of transposon de novo annotation. It will help to understand the research advance quickly and comprehensively in this field, and help readers to select the appropriate transposon annotation tool better and more quickly. However, the manuscript needs to be more concise. For instance, the paragraphs should be reorganized for better reading feeling. In addition, the authors may also need to fully affirm the advantages of the signature-based/library-based methods. At least, the library-based methods are suitable for TE well-annotated species, and have less demanding on the user than the de novo ones.
This article provides a great review of de novo transposon annotation.
Before the article is published, these following cases need to be adjusted:
1: The header of Table 1 is obscured.
2: I hoped that the authors could further clarify the requirement of using these tools and their major applications.
3: Fig. 3 text occlusion problems;
4: For those de novo tools listed in the article, the author should also discuss their shortcomings or limitations so that the reader can fully understand these tools. For example, tools with direct input of NGS or SMS are only suitable for identifying transposons with less diversified sequences and have recently experienced violent replication and proliferation, which don’t aim to obtaining a complete TE reference sequence.
5. It will be useful that the author could briefly expand the relevant content of the post-processing of the annotation results generated by de novo tools. For example, users should distinguish between real transposons and host genes with TE fragments (transposon molecule domestication or multi-copy genes).
Author Response
The paragraphs should be reorganized for better reading feeling.
We have re-read the manuscript and improved the wording and presentation of subjects in several locations. We hope to have addressed the reviewer’s concerns with that.
In addition, the authors may also need to fully affirm the advantages of the signature-based/library based methods.
We interpreted this as a request for a better explanation of advantages signature-based methods have over what we call de novo programs. For that, we have expanded the introduction of the signature-based methodologies section, which already listed several problems de novo programs face. We added:
“Without expectations regarding the structure of mobile elements, these methods also produce many fragmented or overextended TE models. LTR elements are particularly vulnerable to this, and the output of de novo programs may contain (fragments of) solo LTRs, single LTRs with a fragment of an internal sequence on either or both sides, all the way up to LTR-int-LTR-int-LTR structures. Signature-based methods are less susceptible to these particular problems.”
In case a further explanation of the advantage of library-based annotation over direct annotation by the various discovery programs was requested as well, we added an introduction to section 8 (Cleaning up a TE library):
“​​While de novo programs are sometimes used to directly annotate genomes, a careful comparison against a pipeline’s complete library or the combination of the output with previously established models has the advantage that the best match can be chosen between two or more related models. Other advantages of the use of libraries are reproducibility, provenance, and the possibility of incremental improvements”
The header of Table 1 is obscured.
This formatting issue has been corrected.
I hoped that the authors could further clarify the requirements of using these tools and their major applications.
We interpret “tool requirements” to mean computer hardware/software specifications necessary to run the tool. These specifications have been previously published in recent reviews (46,48) and have been purposefully left out here to make room for a more detailed description of the algorithmic approach, input formats, and products that they produce.
We were unsure what the reviewer meant by “major application”, as many of the individual tools represent competitive approaches. General de novo methods are typically applied to complete assemblies or sequence reads - this application is indicated in Table 1 for each tool. Structural-based methods covered in this review are applied to full genome assemblies and therefore we have instead grouped these by the class of element they are designed to detect.
Fig. 3 text occlusion problems;
We were unclear as to which text appears occluded in the reviewer’s copy of the figure, as there wasn’t any overlapping text present in our copy. We did however work to improve this figure as the red “x”s in the “identify composite piles” step confused reviewer #2. We will work with the publisher to make sure that all text is visible in the printed copy. We also added the following text to the figure legend walk the user through what the figure is depicting:
“Figure 3. Self-comparison approaches. These methods attempt an all-vs-all self alignment using the whole assembly or a portion thereof. The self alignments, viewed as a dot plot, will have many off-diagonal alignments representing dispersed similarities. These methods group the alignments into “piles”, defined by their distinct coverage across a region of the assembly. The primary difference between methods is in how they group piles into families. PILER and CARP require that elements are globally alignable, identifying R1/R3 as a distinct family rather than fragments. Grouper and RECON apply single linkage clustering which, in this example, groups all fragments into a single family. RECON further attempts to identify composite families by looking for overrepresented internal edges – in this example the internal edges were not deemed significant (red x’s).”
For those de novo tools listed in the article, the author should also discuss their shortcomings or limitations so that the reader can fully understand these tools. For example, tools with direct input of NGS or SMS are only suitable for identifying transposons with less diversified sequences and have recently experienced violent replication and proliferation, which don’t aim to obtaining a complete TE reference sequence.
We believe that we have addressed the shortcoming of de novo tools in the opening paragraph to section 5: Signature-based methods:
“Purely de novo methods should be able to detect all classes of TE families. However, detecting TEs by sequence repetition alone has the potential to miss low-copy or certainly single-copy members of any well-characterized class of TEs. In addition, these approaches will also include non-TE sequences, such as processed pseudogenes and high-copy gene families.”
And have mentioned throughout the text the difficulties using de novo TE discovery tools in the introduction. A few examples include:
“Over time, class I TEs can change to the point that homology between old and young copies or between copies of two branches is obscure. De novo algorithms will not consider these dissimilar copies to represent the same TE family. This is usually preferable, as a consensus or profile of these aligned dissimilar sequences could be meaningless.”
“Proper clustering of the instances is complicated, even if sometimes aided by apparent bursts of activity of the TE resulting in clear “subfamilies” of instances. None of the de novo programs currently attempt an automated subfamily analysis.”
“... de novo programs are also prone to build in silico chimeras of TEs, in part because integration site preferences of prolific TEs can make them frequently appear in tandem or at the (near) same site of other interspersed repeats.”
We have also discussed the differences between de novo TE discovery tools that rely on either read or assembly data:
“Many de novo methods have been developed that directly operate on next generation sequencing (NGS) or single molecule sequencing (SMS) reads. The primary advantage of this approach is to avoid assembler biases that often cause low-divergence repetitive sequences to be mis-assembled or left out entirely.”
It will be useful that the author could briefly expand the relevant content of the post-processing of the annotation results generated by de novo tools. For example, users should distinguish between real transposons and host genes with TE fragments (transposon molecule domestication or multi-copy genes).
The section “Cleaning up a TE library” is already relatively quite extensive and we have not endeavored to expand on the (many) remaining issues except for the one the reviewer suggested.
The filtering of multi-copy genes from de novo program’s output was already discussed in the paragraph starting with, “Models that represent signature based false positives or segmental duplications can contain coding regions of cellular genes.”
To address the problem of (fragments of) cellular genes occasionally showing up in true TEs we’ve added to the end of that paragraph:
“TEs may carry (fragments of) cellular genes along, so this filter should be conservative.”
Distinguishing domesticated / exapted TE sequence from non-functional TE-derived DNA is not a task de novo programs can or should address. Exaptations can be recognized by measuring the accumulation of mutations, either by comparison to the consensus (if it represents the original TE) or by comparative genomics. Annotating genes that are derived from TEs is appropriate and one of many reasons researchers should not ignore repeat-annotated DNA.
Reviewer 2 Report
The manuscript by Storer et al. provides an overview of the existing methods and software tools for de novo discovery of transposable element (TE) families. The authors summarize the respective results, including the most recent achievements, in a clear and concise fashion, making it easy for a reader to promptly navigate through those. The figures and tables incorporated in the text will further facilitate the task of selecting the most appropriate approach to solving a particular problem. This review should be indeed helpful for the specialists in genomics and the related fields, and, in my opinion, can be accepted for publication in Genes in its present form.
Author Response
N/A
Reviewer 3 Report
An overwhelming variety of programs and algorithms available for transposable elements (TE) annotation makes it difficult for researchers to choose a correct workflow suitable for their task. It makes a review by Storer and co-authors a valuable guide in a field of mobile elements identification, both in theory and practice. The manuscript is very well written and illustrated.
Some minor repetition in the text is expected (given the topic) and it only makes it easier to follow the main points.
Line 3. I would suggest not limiting the approaches to TE discovery reviewed in the manuscript to "newly" sequenced genomes, as they can be useful in a re-evaluation of TE repertoires in some of already explored ones (and even in model species).
Line 30. Please cite an appropriate reference for these estimates
Line 40. Speaking of LINE elements as "close to ideal" markers, it should be mentioned that they were shown to be transmitting horizontally. https://doi.org/10.1186/s13059-018-1456-7
Line 89. A brief overview of an overlap between satellite DNA and interspersed elements could be added here, or at Line 194. For example, this review https://doi.org/10.3390/genes8090230
and more on the topic https://genome.cshlp.org/content/28/5/714
https://www.ncbi.nlm.nih.gov/pmc/articles/PMC6947457/
Line 161. A definition of a term "Model" could be added to the glossary. This term is explained at line 250, but in other parts of the text the meaning may be not so clear to the reader.
Table 1 and 2, Library Generation - "Other" is not explained
Figure 3. It is not clear what are the red crosses at the ends of shorter fragments in RECON piles
Line 348. "Env" explanation is missing; extra comma between H and MT
Line 349. Figure 5. Among SINE-related signatures, conservative A and B boxes of internal Pol III promoter could be mentioned.
Line 388. In addition to CD-HIT and similar tools, a relatively novel (non-greedy) mean shift algorithm was proposed which is promising for a machine-learning based clustering https://www.biorxiv.org/content/10.1101/2022.01.15.476464v1.full.pdf
Figure 6. It would be helpful to provide a reference to Supplementary table here.
Line 656. "A exact" -> "An exact"
Author Response
Line 3. I would suggest not limiting the approaches to TE discovery reviewed in the manuscript to “newly” sequenced genomes, as they can be useful in a re-evaluation of TE repertoires in some of already explored ones (and even in model species).
We have removed this clause from the title of the manuscript.
Line 30. Please cite an appropriate reference for these estimates
Since the statement was meant to emphasize the large proportion of TE-derived DNA that is ancient, we left out the estimated overall coverage of the human genome and added references to the remaining statement.
Line 40. Speaking of LINE elements as “close to ideal” markers, it should be mentioned that they were shown to be transmitting horizontally. https://doi.org/10.1186/s13059-018-1456-7
Horizontal transfer is not an issue when applying the standard and preferred method of looking at the presence or absence of copies at individual sites. It would be a problem if one just checks for the general presence of copies of a (very similar) TE in different genomes, but once you have genome sequence for both, that strategy is obsolete.
Line 89. A brief overview of an overlap between satellite DNA and interspersed elements could be added here, or at Line 194. For example, this review https://doi.org/10.3390/genes8090230
and more on the topic https://genome.cshlp.org/content/28/5/714
https://www.ncbi.nlm.nih.gov/pmc/articles/PMC6947457/
A good point, which we now (very briefly) address by adding the following text to the section regarding co-duplication of TEs by other mechanisms a paragraph below line 194:
“Perhaps in part due to the frequency of tandem copies mentioned above, TE instances appear overrepresented in satellite-like tandem repeats (59).”
Reference 59 is the suggested 2017 review published in “Genes”.
Line 161. A definition of a term “Model” could be added to the glossary. This term is explained at line 250, but in other parts of the text the meaning may be not so clear to the reader.
Specifically, we are referring to sequence models here (e.g. consensus or pHMM). To introduce what we mean by model we have changed the preceding sentence to:
“This is usually preferable, as a consensus or profile HMM sequence model of these aligned yet dissimilar sequences could be meaningless.”
And added the following to the glossary:
“Sequence Model (n): A summary representation of a set of nucleotide or protein sequences, typically a consensus sequence, a sequence profile, or a profile Hidden Markov Model.”
Table 1 and 2, Library Generation - “Other” is not explained
This column indicates that the tool generates a library in a format other than consensus sequences or exemplars, as opposed to foregoing library generation altogether. We have changed the text to make this clearer:
“In Table 1 de novo tools are characterized by the level of granularity they produce (families, instances, other), and by the types of family models produced (consensi, exemplars, or other tool-specific representations).”
Figure 3. It is not clear what are the red crosses at the ends of shorter fragments in RECON piles
The red “x”s are indicating alignment endpoints that were considered by RECON’s composite detection algorithm and, in this example, determined not to be significant. To make this clearer we have enlarged this section of the figure somewhat and changed the figure legend to read:
“Figure 3. Self-comparison approaches. These methods attempt an all-vs-all self alignment using the whole assembly or a portion thereof. The self alignments, viewed as a dot plot, will have many off-diagonal alignments representing dispersed similarities. These methods group the alignments into “piles”, defined by their distinct coverage across a region of the assembly. The primary difference between methods is in how they group piles into families. PILER and CARP require that elements are globally alignable, identifying R1/R3 as a distinct family rather than fragments. Grouper and RECON apply single linkage clustering which, in this example, groups all fragments into a single family. RECON further attempts to identify composite families by looking for overrepresented internal edges – in this example the internal edges were not deemed significant (red x’s).”
Line 348. “Env” explanation is missing; extra comma between H and MT
We have removed the extra comma.
Line 349. Figure 5. Among SINE-related signatures, conservative A and B boxes of internal Pol III promoter could be mentioned.
We agree with the reviewer, and have edited Figure 5 to include SINE-related signatures. In addition, we have also modified the figure caption to read:
Figure 5: Examples of commonly-used TE Signatures for Detection. Structural features: the identification of LTR/ERV elements, class II elements, non-LTR retrotransposable elements, and Helitrons can be achieved by searching for LTRs (~100-1000bp direct repeats), TIRs (~10-40bp inverted repeats), TSDs (6-21bp on average duplications), and hairpin structures, respectively. In addition, the A and B boxes seen in RNA polymerase III promoters and 3’ terminal A/T-rich sequence can be used to identify SINE elements. Motifs/Protein Homology: the order, orientation, and similarity to protein domains is key to homology-based searches. Gag: group specific antigen; PR: pathogenesis-related; RT: reverse transcriptase; EN: endonuclease; Env: envelope; RH: ribonuclease H; MT: methyltransferase; YR: tyrosine recombinase. Other sequence structures (not seen in the figure above) observed in LINEs poly-A or simple-repeat tails, the RT and apurinic–apyrimidinic EN (APE) domains of the Pol protein.
Line 388. In addition to CD-HIT and similar tools, a relatively novel (non-greedy) mean shift algorithm was proposed which is promising for a machine-learning based clustering https://www.biorxiv.org/content/10.1101/2022.01.15.476464v1.full.pdf
There is much work going on in this area and there is much room for improvement in the methods employed to cluster TE instances into families and further to identify redundancy in TE libraries. To point this out we have added the following sentence to the end of that paragraph:
“This is an area that is likely to see improvement in coming years as novel sequence distance estimation (https://doi.org/10.1101/2020.11.13.381814) and clustering techniques (https://doi.org/10.1101/2022.01.15.476464) are evaluated in the context of TE families.”
Figure 6. It would be helpful to provide a reference to Supplementary table here.
We have added a reference to this table in the figure legend.
Line 656. “A exact” should be “An exact”
Fixed.